# Performance Optimization of Nitrogen Dioxide Gas Sensor Based on Pd-AlGaN/GaN HEMTs by Gate Bias Modulation

**DOI:** 10.3390/mi12040400

**Published:** 2021-04-05

**Authors:** Van Cuong Nguyen, Kwangeun Kim, Hyungtak Kim

**Affiliations:** 1School of Electronic and Electrical Engineering, Hongik University, Seoul 04066, Korea; nvcuong@mail.hongik.ac.kr; 2School of Electronics and Information Engineering, Korea Aerospace University, Gyeonggi 10540, Korea; kke@kau.ac.kr

**Keywords:** gate bias modulation, palladium catalyst, gallium nitride, nitrogen dioxide gas sensor, high electron mobility transistor

## Abstract

We investigated the sensing characteristics of NO_2_ gas sensors based on Pd-AlGaN/GaN high electron mobility transistors (HEMTs) at high temperatures. In this paper, we demonstrated the optimization of the sensing performance by the gate bias, which exhibited the advantage of the FET-type sensors compared to the diode-type ones. When the sensor was biased near the threshold voltage, the electron density in the channel showed a relatively larger change with a response to the gas exposure and demonstrated a significant improvement in the sensitivity. At 300 °C under 100 ppm concentration, the sensor’s sensitivities were 26.7% and 91.6%, while the response times were 32 and 9 s at *V*_G_ = 0 V and *V*_G_ = −1 V, respectively. The sensor demonstrated the stable repeatability regardless of the gate voltage at a high temperature.

## 1. Introduction

Nanotechnology is being employed in the research of sensor devices to a great extent. Owing to the unique properties of the nanomaterials such as great adsorptive capacity due to the large surface-to-volume ratio, the ability of tuning electrical properties by controlling the composition and the size of the nanomaterial, the easy configuration and integration in low-power microelectronic systems, the gas sensors based on nanotechnology are excellent candidates for sensitive detection of chemical and biological species [1,2,3,4]. 

Nitrogen dioxide (NO_2_) is one of the most harmful gases released into the atmosphere from both natural sources and human activities. Prolonged exposure with a high concentration of NO_2_ can cause lung tissue inflammation, aggravate respiratory diseases, particularly asthma, leading to respiratory symptoms. The LC50 (the lethal concentration for 50% of those exposed) for a 1-h exposure of NO_2_ for humans has been estimated to be 174 ppm [5]. In industry, nitrogen dioxide gas is emitted from the burning of fuel, the exhaust of furnaces and power plants, etc. Therefore, the development of ppm-level NO_2_ gas sensors that can operate at very high temperatures is necessary. 

Gas sensors based on metal oxide have been intensively investigated for several decades [6,7,8,9]. Despite high sensitivity and easy fabrication, they are not considered a promising candidate for extreme environment electronics due to a long response time, poor selectivity towards any specific gas, and unstable operation at harsh environmental conditions [10]. To develop a gas sensor that operates at high temperatures, gallium nitride-based gas sensors [11,12,13,14,15,16,17] recently have attracted great attention thanks to the wide bandgap of 3.4 eV, high thermal, and chemical stability. 

Currently, NO_2_ gas sensors based on AlGaN/GaN high electron mobility transistors (HEMTs) have been the subject of intense research [18,19,20,21,22,23,24]. The large variation of the sensitivity of these sensors was explained by the trade-off between the sensitivity (*S*) and the base drain current (*I*_0_). For example, a Pt-AlGaN/GaN sensor showed a significant current change of 2.8 mA measured at 400 °C under 450 ppm NO_2_, but the high base current of 16.5 mA led to 17% of sensitivity [21]. Another Pt-AlGaN/GaN sensor showed 5.5% of sensitivity while it had a high current change of 1.801 mA and a high base current of 33 mA [22]. On the other hand, a NH_3_ gas sensor archived an ultra-high sensitivity of 18,300% at 150 °C, but the base current was limited in the pA range [25].

There are some approaches to optimize the sensitivity of sensors based on HEMTs. Firstly, choosing an appropriate catalyst is very important. Since the sensors work based on the chemical mechanism, the catalyst directly affects the sensor characteristics such as working temperature, sensitivity, response, and recovery times. Our previous work proved that sensors based on Pd-AlGaN/GaN HEMT showed better performance than the Pt one [24]. Second, the surface treatment, like hydrogen peroxide treatment [26] or plasma treatment [27], can improve the gas sensor’s sensitivity. Third, since HEMTs are 3-terminal devices, they provide the possibility of adjusting the gate voltage to control the drain current, thereby optimizing the sensor’s performance, which is a superior advantage when compared to diode-type sensors. 

In this paper, we comprehensively investigated the sensitivity enhancement of the NO_2_ gas sensor based on Pd-AlGaN/GaN HEMT by gate bias modulation. When the sensor was biased close to the threshold voltage, the sensitivity and the response time were much improved. The Technology Computer-Aided Design (TCAD) simulation revealed that the improvement of sensitivity was attributed to the larger change in the channel electron concentration near the threshold.

## 2. Materials and Methods

The fabrication process of the sensor is shown in Figure 1. AlGaN/GaN HEMT-type sensors were fabricated at the Inter-University Semiconductor Research Center (ISRC), Seoul, Korea. The AlGaN/GaN-on-Si substrate consisted of a 10 nm in situ SiN_x_ layer, a 13 nm Al_0.3_Ga_0.7_N barrier layer, a 4.2 μm i-GaN layer, and AlGaN/AlN buffer layers. AlGaN layer parameters such as Al-content and the thickness were optimized for the maximum transconductance by increasing Al-content and thinning the thickness. The source and drain contacts with Ti/Al/Ni/Au (20/120/25/50 nm) were formed by e-beam evaporation with a lift-off process and followed by rapid thermal annealing (RTA) at 830 °C for 30 s in N_2_ ambient. Then, 300 nm-depth mesa isolation was formed by inductively coupled plasma (ICP) etching with BCl_3_/Cl_2_ to define the active region. The 30 nm Pd layer as a gate electrode was then formed by e-beam evaporation and a lift-off process. Afterward, the interconnect bi-layer probing pads of Ti/Au with thickness 20/300 nm were formed by e-beam evaporation and lift-off. A passivation layer of 200 nm SiN_x_ was deposited using plasma-enhanced chemical vapor deposition (PECVD) at 190 °C in order to protect the sensor’s surface. Finally, the SiN_x_ layer was patterned and etched to open the Pd-gate to the ambient and the contact pads for measurement. The dimensions of the Pd-gate electrode were 24 µm × 120 µm, the source-gate and gate-drain spacings were 2 µm. The fabricated AlGaN/GaN HEMT exhibited sheet resistance, 2DEG mobility, and sheet carrier density of 493 Ω/□, 1420 cm^2^/(V⋅s), and 8.9 × 10^12^ cm^−2^, respectively. The microscope image of the fabricated Pd-AlGaN/GaN HEMT device is shown in Figure 2a.

Gas sources consist of synthesized dry air as the background gas, and 100 ppm NO_2_ as target gas. The background and the target gases were mixed by mass flow controllers (MFCs) to archive the different concentrations of NO_2_. The combined total gas flow was set to 200 sccm in all measurements. The sensors were loaded in a chamber containing a hot chuck to control the operating temperature. The DC and transient characteristics of the sensor were measured using the HP 4155A semiconductor parameter analyzer.

The sensitivity is defined as the ratio between the change of drain current and initial current: (1)$$Sensitivity(\%)=\frac{I_{0}-I_{NO2}}{I_{0}}=\frac{\Delta I}{I_{0}},$$
where *I*_0_ and *I*_NO2_ are drain currents under the flow of dry air and NO_2_ gas, respectively. The response and recovery times were calculated from transient characteristic, where they showed 90% of the total change (Δ*I*) in drain current [28]. 

The sensing mechanism of nitrogen dioxide sensors based on Pd-AlGaN/GaN was investigated and reported in our previous work [24]. When nitrogen dioxide gas is adsorbed on the Pd catalyst layer, the nitrogen dioxide molecules are dissociated in nitrogen monoxide (NO) going to the gas phase and oxygen ions on the Pd surface [29,30]. Then, the negatively charged oxygen ions diffuse through the gate and reach the surface of AlGaN layers. Here, they affect the number of mobile carriers in two-dimensional electron gas (2DEG) of the HEMT structure, leading to a reduction of drain current (Figure 2b).

## 3. Results

The transfer characteristics of the fabricated HEMT at different temperatures were shown in Figure 3. The device exhibited stable operations up to 300 °C due to GaN’s thermal stability.

Figure 4 showed the response of the sensor at different gate voltages at different biases and temperatures. While the adsorption of oxygen ions on the Pd/AlGaN interface occurs even at room temperature, the desorption totally takes place at high temperatures beyond 200 °C, which directly affects the recovery of the sensor. This indicates that this sensor is more suitable for high temperature operation. At a higher temperature, the adsorption and desorption of negatively charged oxygen ions took place more efficiently, resulting in faster response and recovery times. Starting from 300 °C, the sensor fully recovered to the base current, which agreed with other studies [21,22,23]. Additionally, the sensitivity was improved at higher temperatures (Figure 4d). When the device is biased closed to the threshold voltage, the reduction of the current level leads to a higher sensitivity, which is similar to the gate recess approach [31]. Since the gate bias modulation is a damage-free technique, it exhibited the advantage over the gate recess method to improve the sensor’s performance.

To discover the physical mechanism of the sensitivity improvement, the simulation on Silvaco TCAD was performed with our wafer parameters. The gas injection was simulated by the variation of the gate work function because NO_2_ incorporation through Pd-catalyst eventually introduces oxygen ions onto AlGaN barrier surface. The oxygen ions provided a negative charge, leading to an increase of the Pd work function. The simulation results showed that when the device was biased at −1 V, the difference of the conduction band and electron quasi-Fermi level was much smaller than that at 0 V (Figure 5a,b), which resulted in a remarkable reduction of channel electron density (Figure 5c). Relative change of electron concentration responding to the gas exposure was increased from 14.5% (*V*_G_ = 0 V) to 30% (*V*_G_ = −1 V).

When HEMT sensor is biased at a high negative gate voltage, the drain current became too low, which may affect the stability of sensing characteristics. Our sensor showed good repeatability regardless of the gate voltage. In 10 continuous cycles at 300 °C, the sensor totally recovered to the initial current, even at a high negative gate voltage of −1 V, when the drain current was limited in the μA range (Figure 6a). There was an improvement in response time, from 32 s (*V*_G_ = 0 V) to 9 s (*V*_G_ = −1 V), while the recovery times were 36 and 48 s, respectively. The repeatable response of the sensor indicated that the gate bias modulation did not interact with the sensing mechanism. The sensor also exhibited good stability in time. No significant change in transfer characteristics was observed when the measurement was done as-fabricated sensor and 6 months later (Figure 6b).

Figure 7a–d showed the response of the sensor under different concentrations of NO_2_ from 10 to 100 ppm at different gate voltages from 0 to −1 V at 300 °C. The sensor exhibited a huge improvement in sensitivity for all concentrations of NO_2_ (Figure 7e). The sensitivity under 10 ppm of NO_2_ was 6% at *V*_G_ = 0 V and 45.4% at *V*_G_ = −1 V, which is much improved when compared to other sensors based on AlGaN/GaN HEMTs (Table 1). The improvement of response time at lower gate voltages was also recognized (Figure 7f). 

## 4. Conclusions

We demonstrated the optimization of NO_2_ gas sensor’s performance by adjusting the gate bias of NO_2_ gas sensor based on Pd-AlGaN/GaN HEMTs, which is a superior advantage compared to Schottky diode-type sensors. The sensor exhibited a notable improvement of the sensitivity and response time when biased at −1 V. The physical mechanism of this phenomenon was explained by the reduction of channel electron density in TCAD simulation.

## Figures and Tables

**Figure 1 micromachines-12-00400-f001:**
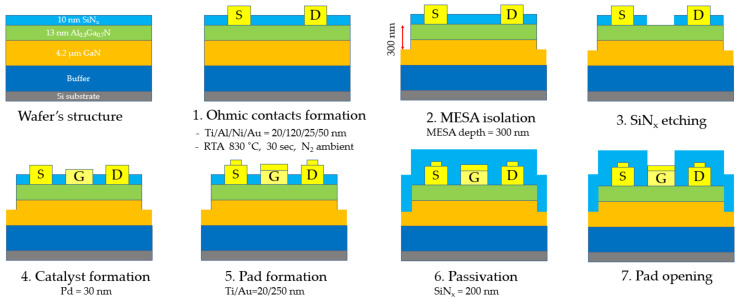
Fabrication process of Pd-AlGaN/GaN high electron mobility transistors (HEMT) sensor.

**Figure 2 micromachines-12-00400-f002:**
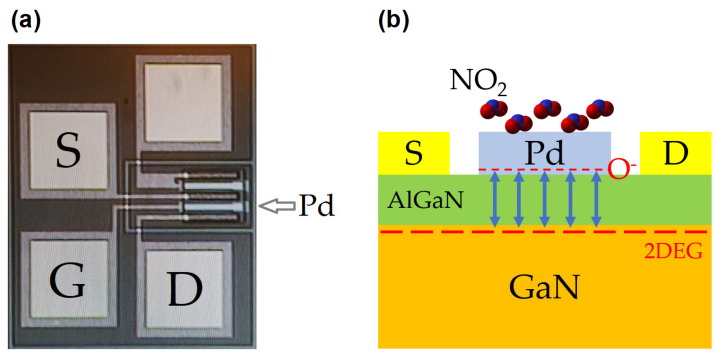
Microscope image of fabricated Pd-AlGaN/GaN HEMT device (**a**), and the sensing mechanism of the sensor (**b**).

**Figure 3 micromachines-12-00400-f003:**
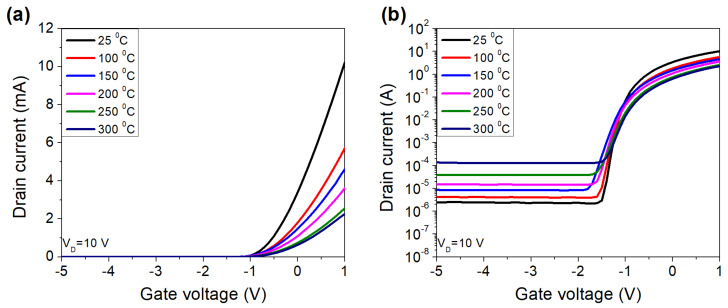
Transfer characteristics of fabricated device in linear (**a**) and log (**b**) scales.

**Figure 4 micromachines-12-00400-f004:**
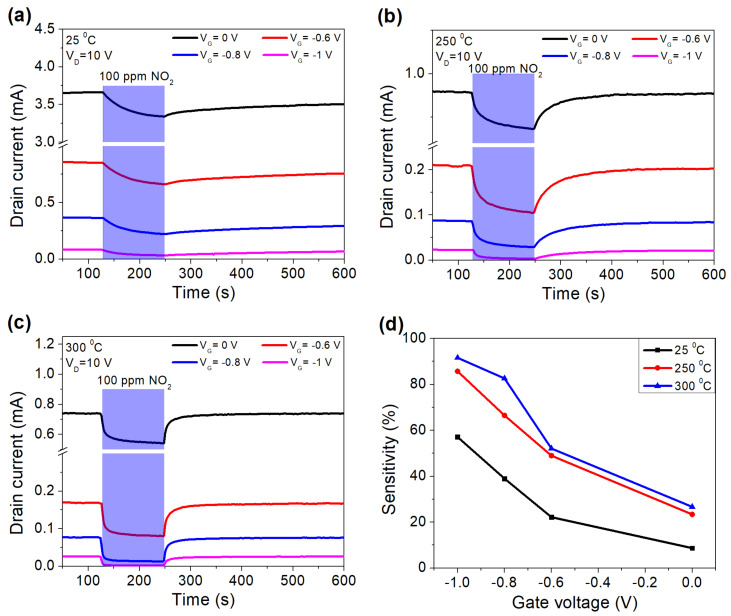
Sensor’s response at 25 °C (**a**), 250 °C (**b**), 300 °C (**c**), at different gate biases and sensitivity as a function of gate voltage (**d**).

**Figure 5 micromachines-12-00400-f005:**
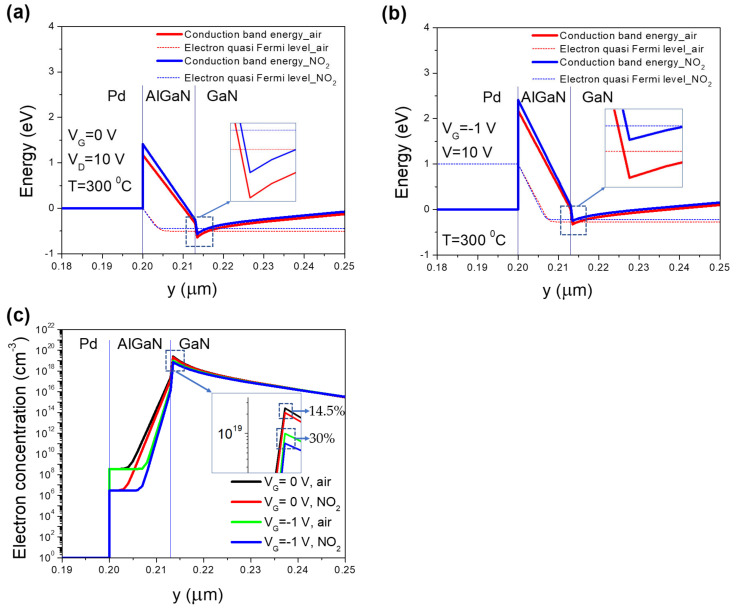
Silvaco TCAD simulation: (**a**) conduction band energy at *V*_G_ = 0 V and (**b**) *V*_G_ = −1 V, (**c**) electron concentration.

**Figure 6 micromachines-12-00400-f006:**
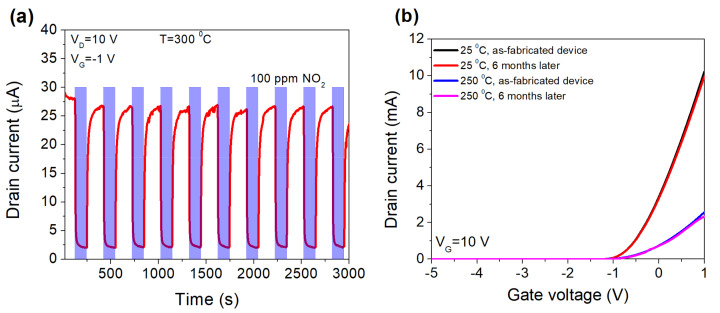
Repeatability of the sensor’s response at gate voltages −1 V (**a**) at 300 °C and the stability of the sensor (**b**).

**Figure 7 micromachines-12-00400-f007:**
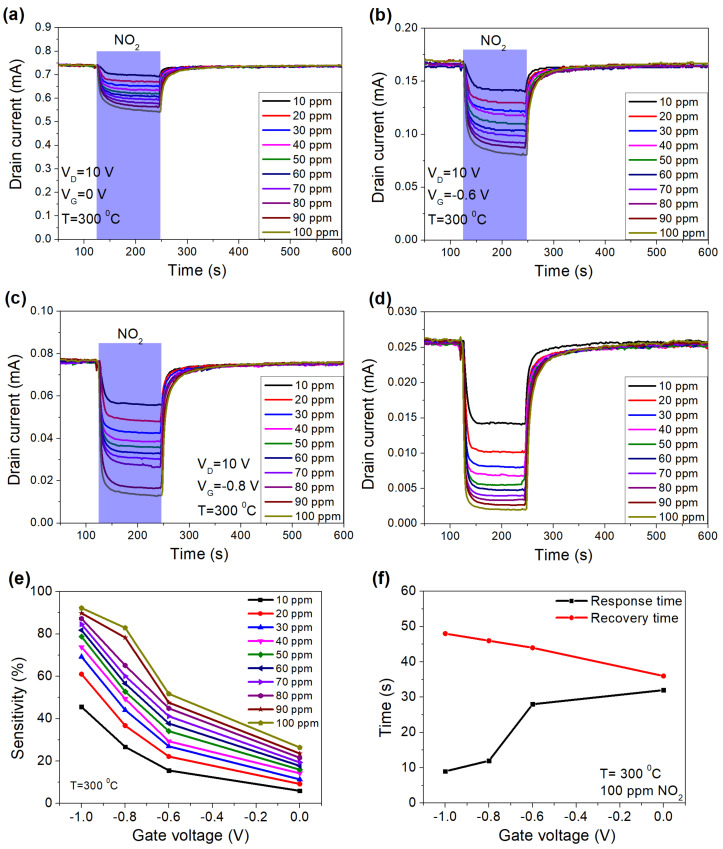
Response of the sensor under the different concentrations of NO_2_ at different gate voltages of 0 V (**a**), −0.6 V (**b**), −0.8 V (**c**), −1 V (**d**), sensitivity at different concentrations (**e**), the response and recovery times as a function of gate voltage (**f**), at 300 °C.

**Table 1 micromachines-12-00400-t001:** The comparison of key parameters of NO_2_ gas sensor.

Sensing Materials	Structure	NO_2_ (ppm)	*T* (°C)	*I*_0_ (mA)	Sensitivity (*S*)/ Response (*R*)	Response Time (s)	Recovery Time (s)	References
Pt	HEMT (*V*_G_ = 0 V)	10	300	35	S = 1%	~4 min	~4 min	[21]
Pt	HEMT (floating gate)	10	300	33	S = 5.5%	~2 min	~5 min	[22]
Pt	HEMT (*V*_G_ = 0 V)	100	300	43.8	S = 7%	~3 min	~2 min	[23]
SnS_2_/RGO	3D	8	RT	-	R = 49.8%	153	76	[32]
F-SWCNTs	Thin film	50	RT	-	S = 37%	4 min	~8 min	[33]
MoS_2_ p-n junction	Thin film	20	RT	-	R ≈ 90	150	30	[34]
Pt	HEMT (floating gate)	10	275	33.9	S = 5.1%	56	285	[35]
Pd	HEMT (*V*_G_ = 0 V)	10	300	0.74	S = 6%	32	36	This work
Pd	HEMT (*V*_G_ = −1 V)	10	300	26 μA	S = 45.4%	9	48	This work

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
