# Peer review of "Performance Optimization of Nitrogen Dioxide Gas Sensor Based on Pd-AlGaN/GaN HEMTs by Gate Bias Modulation"

_micromachines, 2021, doi:10.3390/mi12040400_

Round 1
Reviewer 1 Report
Nguyen et. al investigated pd-AlGaN/GaN HEMT for sensing nitrogen dioxide in elevated temperature and optimising the performance by modulating gate bias. The paper is well presented with solid experimental results and the observations are soundly supported by TCAD simulation. As NO2 sensing is an important industrial safety application, this work would be of interest to the community. I recommend to publish this paper as it is in the present form.
Author Response
Thank you for your time for review.
Reviewer 2 Report
The manuscript submitted by author related to comprehensively investigated the sensitivity enhancement of the NO2 gas sensor based on Pd-AlGaN/GaN HEMT by gate bias modulation and when the sensor was biased close to the threshold voltage, the sensitivity and the response time were much improved are the good efforts made by authors. The manuscript is within the scope of this journal.
The manuscript can be accepted for publication after the revision. Few of the suggestion are listed below for authors attention.
- Under the introduction section the very first paragraph “The gas sensors based on GaN have been intensively studied during the last decades [1-6]. Thanks to such superior properties when compared to silicon as high saturation voltages, high thermal stability, and wide bandgap of 3.4 eV [7-9], the GaN material has been a promising candidate for extreme environment electronics.” Looks very irrelevant. It needs to be removed from the beginning and need to be incorporated somewhere inside this section.
- The introduction section is very poorly witten cannot be accepted in present form. This section needs to be rewritten again. During writing this section try to implement the following points in this section.
- Why there is need to develop NO2 gas sensor, problem statements, Current chemical sensors for NO2 detection based on metal oxide nanomaterials need to be review briefly.
- Start the introduction with some following paragraphs. But this paragraph needs to be paraphrase’s before using in the script. “Nanotechnology is enabling the production of efficient sensors with broad range of applications. The unique properties of the nanomaterials make them suitable candidates for sensitive detection of chemical and biological species because they exhibit great adsorptive capacity due to the large surface-to-volume ratio, produce great modulation of the electrical signal upon exposer to analytes due to the great interaction zone over the cross sectional area (Debye length), enable tuning electrical properties by controlling the composition and the size of the nanomaterial, and ease configuration and integration in low-power microelectronic systems[1,2]”. And provide relevant refrences. Few references are provided below for adding. [Journal of Science: Advanced Materials and Devices 1, 431-453, 2016; ACS Sensors 1, 55-62, 2016].
- In page 2, Figure 1. Fabrication process of Pd-AlGaN/GaN HEMT sensor is not added on right place. Please check it.
- How the authors have performed the fabrication process of Pd-AlGaN/GaN HEMT sensor. Is they have follow any protocol from literature or self-adopted? Need clarification.
- I can see the response of the sensor at different gate voltages at different biases and temperatures. But no change in NO2 concentration (100ppm) was noticed. How the different NO2 concentration will impact the sensitivity of sensor need to be study.
- Provide with the experimental sensor set up details of NO2 sensing by AlGaN/GaN HEMTs.
- How the author maintains different concentrations of NO2?
- How was the response and recovery time of the sensor? Need to be address.
- Stability of the sensor also need to be study.
- Results and discussion need to be elaborated more.
- English of the script need the help of native speaker to reduce the grammatical and other typos error.
- Mechanism for the NO2 sensing is missing. Authors should provide with plausible schematic mechanism for NO2
Reviewer 3 Report
NO2 sensing is an interesting and important work. It is worth writing an article about it. But I have certain doubts/ enquiry about the article.
1- The following reports have been available on the Pd-AlGaN/GaN HEMTs based hydrogen sensor.
https://doi.org/10.1109/JSEN.2013.2243430
10.1109/ICSENS.2012.6411311
In present article, it has been reported that mentioned material is suitable for NO2 sensing. So, I have doubt on selectivity of as-developed sensor. Did author check the selectivity of sensor? Did author tried to analysed as-developed sensor with other gases.
2- It seems that the similar kind of work (NO2 sensing based on Pd-AlGaN/GaN HEMTs), author already reported in following articles:
https://scholarworks.bwise.kr/hongik/handle/2020.sw.hongik/1969
https://doi.org/10.5573/JSTS.2020.20.2.170
So, what is the difference between reported work and present work? What is the novelty of present article?
Round 2
Reviewer 2 Report
The authors have satisfactory revised the script.
Author Response
Thank you for your review and suggestion.
Reviewer 3 Report
The answers given by authors are satisfactory. I am in favour the acceptance of manuscript after following minor corrections:
1- Author used different formats for defining the values such as “while the response times were 15 32 s and 9 s at VG= 0 V and VG=-1 V, respectively” ,
“recently have attracted great attention thanks to the wide 41 bandgap of 3.4V, high thermal and chemical stability.”
And “a 72 13-nm Al0.3Ga0.7N barrier layer”.
It is clearly observable that some place author used space between the numeric term and unit (for example VG= 0V), some place no space is given (for example 3.4V) and some place there is ‘-‘ used (for example 13-nm). I would like to suggest author to follow same pattern for defining the units for entire manuscript.
2- In the caption of figure 1, author used term “wafer’s parameters”, I think it should be “Wafer’s structure”.
3- There is a unit mistake in line number 88.
“density of 493 Ω/□, 1420 cm2/(V.s) and 8.9×1012 cm−2, respectively.”
Provide proper unit in the “□”.
4- The title of the graph’s axes found some places in bold and some places in normal text. To make the manuscript more attractive, I would like to suggest the author to make it uniform.
5- The references provided for performance comparison in Table 1 are very less. There is no purpose of comparison with only three studies. Author should include more latest- references for comparison.
6- To make manuscript more suitable for readers, I would like to suggest the author make the table generalized for operating temperature instead of 300˚ Also, NO2 sensors based on other materials can be included for comparison. Author may use following latest references:
i. Flexible, 3D SnS2/Reduced graphene oxide heterostructured NO2 sensor (https://doi.org/10.1016/j.snb.2019.127445)
ii. Thin film chemiresistive gas sensor on single-walled carbon nanotubes-functionalized with polyethylenimine (PEI) for NO2 gas sensing (https://doi.org/10.1007/s12034-020-2043-6)
iii. MoS2 Van der Waals p–n Junctions Enabling Highly Selective Room‐Temperature NO2 Sensor (https://doi.org/10.1002/adfm.202000435)
iv. Enhanced NO2 Gas Sensing Performance of Multigate Pt/AlGaN/GaN High Electron Mobility Transistors (https://doi.org/10.1149/1945-7111/abed42)
The sensitivity of a sensor can not be the only parameter for comparison. It is strongly recommended to include in the table some more columns such as “Response time”, “Recovery Time”, “Operating temperature” etc.
7- In the caption of figure 2(a) author wrote “definition of response and recovery times” but it is a very confusing statement. I would like to advise the author to write a proper explanation in figure caption.
8- The recovery of the sensor is natural process or any external energy source such as UV light has been used for recovery of the sensor?
